# NOCTIS: Novel Object Cyclic Threshold based Instance Segmentation

## Abstract

Instance segmentation of novel objects instances in RGB images, given some example images for each object, is a well known problem in computer vision. Designing a model general enough to be employed for all kinds of novel objects without (re-) training has proven to be a difficult task. To handle this, we present a new training-free framework, called: Novel Object Cyclic Threshold based Instance Segmentation (NOCTIS). NOCTIS integrates two pre-trained models: Grounded-SAM 2 for object proposals with precise bounding boxes and corresponding segmentation masks; and DINOv2 for robust class and patch embeddings, due to its zero-shot capabilities. Internally, the proposal-object matching is realized by determining an object matching score based on the similarity of the class embeddings and the average maximum similarity of the patch embeddings with a new *cyclic* thresholding (CT) mechanism that mitigates unstable matches caused by repetitive textures or visually similar patterns. Beyond CT, NOCTIS introduces: (i) an appearance score that is unaffected by object selection bias; (ii) the usage of the average confidence of the proposals' bounding box and mask as a scoring component; and (iii) an RGB-only pipeline that performs even better than RGB-D ones. We empirically show that NOCTIS, without further training/fine tuning, attains state-of-the-art results regarding the mean AP score, w.r.t. the best RGB and RGB-D methods on the seven core datasets of the BOP 2023 challenge for the "Model-based 2D segmentation of unseen objects" task. [1]

## 1 Introduction

Instance segmentation (and detection) is one of the key problems in robot perception and augmented reality applications, since it tries to identify and locate object instances in images or videos via bounding boxes and segmentation masks; e.g. a robot wants to identify a specific object instance on a conveyor belt. Classical supervised deep learning methodologies such as Ren et al. (2015); He et al. (2017); Lin et al. (2020); Labbé et al. (2020); Su et al. (2022) have achieved good performances, however, for each target object they require an expensive (re-) training or, at least, a fine-tuning step. Hence, most of the time, adding novel/unseen objects begs for additional training data, either synthetic or real, with (human-made) annotations. As a consequence, the usage of said supervised methods for industry, especially for short-time prototype development cycles, where the target object changes constantly, is unfeasible.

In this work, we introduce NOCTIS, a fully training-free RGB-only framework, for novel objects instance segmentation that achieves state-of-the-art (SOTA) performances on the BOP 2023 core datasets. NOCTIS builds upon the strengths of two recent pre-trained models, Grounded-SAM 2 (Ren et al., 2024b) (GSAM 2) for precise mask proposal generation and DINOv2 (Oquab et al., 2024) for robust visual descriptors, but departs from previous approaches like CNOS (Nguyen et al., 2023); NIDS-Net (Lu et al., 2025) and SAM-6D(Lin et al., 2024) in several critical ways.

First, we introduce an appearance score that is unaffected by object selection bias, differently from the appearance-based score previously introduced in SAM-6D, via evaluating it over all templates per object and aggregating results, rather than relying on the single template with the highest semantic score across all templates from all objects. This removes the strong bias towards one object–template pair and leads to a more consistent detection quality.

---

[1] Code available at: Only at publication

Second, we propose a *cyclic* thresholding (CT) mechanism, a novel patch-filtering strategy designed to handle repetitive textures and visually similar patterns that can lead to many-to-one matches. Unlike nearest neighbor matching (e.g. Simakov et al. (2008); Oron et al. (2018)), CT relaxes strict mutual matching requirements, tolerating some distance between a patch and its *cyclic*/round-trip patch, while filtering out unreasonable ones (see Section 3.3). This yields a more reliable proposal-template matching and improves the appearance-based scoring process.

Third, we are the first, to the best of our knowledge, to incorporate mask and bounding-box confidence values, readily available from modern mask proposal generators, into the final object matching score. While such confidence measures are commonly produced by detection and segmentation models (e.g. Grounding-DINO (Liu et al., 2024); SAM (Kirillov et al., 2023); FastSAM (Zhao et al., 2023); Grounded-SAM (Ren et al., 2024b) (GSAM)), they have not been exploited in this task before, and we demonstrate their positive impact through ablation studies.

Finally, we test our approach on the seven core datasets of the BOP 2023 challenge (Hodan et al., 2024) for the "2D instance segmentation of unseen objects" task; and we show that our method NOCTIS, without further training, performs better than other RGB and RGB-D methods in terms of the Average Precision (AP) metric; challenging the assumption that depth information is required for top-tier performances in this domain. Moreover, we surpass the best published method NIDS-Net (Lu et al., 2025) by a significant margin of absolute $3.4\%$ mean AP; while for the unpublished ones, we are on par with the (updated) best one and overcoming the second best one by $0.8\%$.

Our contributions can be summarized as follows:

- We propose NOCTIS, an RGB-only zero-shot novel objects instance segmentation framework that uses foundation models and performs on par or better than the SOTA ones.

- An unbiased appearance score that aggregates over all templates to remove selection bias.

- A novel cyclic thresholding mechanism for robust patch matching to mitigate matching instability from repetitive textures.

- Inclusion of the proposal's confidence as a weight for the object matching score.

## 2 RELATED WORK

**Pre-trained models for visual features**    The usage of pre-trained foundation models has become pervasive due to their strong performance across diverse downstream tasks. Notably, research efforts such as Visual Transformers (ViT) (Dosovitskiy et al., 2021); CLIP (Radford et al., 2021); DINOv2 and others (Caron et al., 2021; Cherti et al., 2023); have focused on large-scale image representation learning to improve generalization. These models encapsulate extensive visual knowledge, making them suitable as backbones for a variety of tasks, including image classification, video understanding, depth estimation, semantic segmentation, and novel instance retrieval. The main challenge, however, lies in harnessing their capabilities effectively for a specific target domain. We adopt DINOv2 as our feature extractor, leveraging its ability to produce high-quality and robust descriptors for previously unseen instances.

**Segment anything**    Another area where foundation models currently excel is image segmentation/semantic mask generation, with Segment Anything (SAM), a ViT-based model, being the forerunner. Since SAM can be computationally demanding, several variations have been proposed for real-world scenarios to reduce costs by replacing components with smaller (less parameters/weights) ViT models (Zhang et al., 2023; Ke et al., 2023), or even CNN-based ones (Zhao et al., 2023; Zhou et al., 2024; Wang et al., 2024). Notably, its successor, SAM 2 (Ravi et al., 2025), while having some additional features like video tracking, achieves a higher quality in terms of Mean Intersection over Union (mIoU) than SAM and is also more efficient computation- and memory-wise. In recent times, it has become an established practice to combine the strengths of multiple models in a modular way to solve complex problems. Indeed, a standard practice to tackle segmentation problems consists of combining open-set object detectors (Li et al., 2022; Jiang et al., 2024; Ren et al., 2024a; Liu et al., 2024) with a SAM variant. We adopt GSAM 2 in this modular spirit but extend its utility beyond simple mask generation by incorporating its bounding box and mask confidence values directly into our scoring framework; an element absent in prior work.

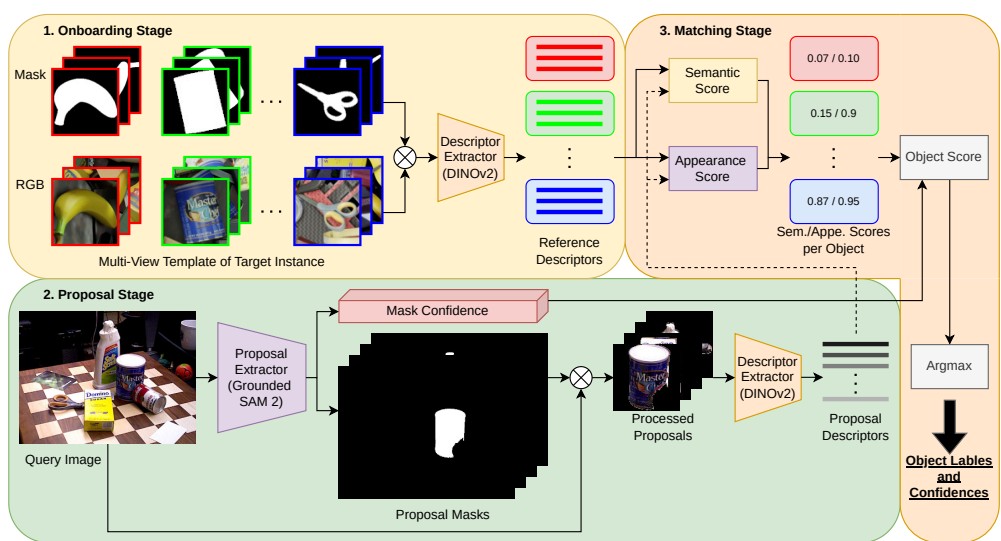

Figure 1: The three NOCTIS stages: onboarding stage, represents each object via descriptors from templates (Section 3.1); proposal stage (Section 3.2), where proposals (masks) and their descriptors from the query RGB image are generated; lastly, in the matching stage, object labels and confidences are assigned to each proposal based on their descriptors (Section 3.3).

**Segmentation of unseen objects** Traditionally, instance segmentation methods, like Mask R-CNN (He et al., 2017) or similar (Ren et al., 2015; Lin et al., 2020; Su et al., 2022), used to be fine-tuned on specific target objects (Sundermeyer et al., 2023). Even though these methods have been demonstrated to be robust in challenging scenarios with heavy occlusions and lighting conditions, they lack the flexibility to handle novel objects without retraining, known as a "closed-world" setting; a limitation that significantly hinders their applicability in real-world settings. To overcome this limitation, progress was made in the task of novel object instance segmentation, where ZeroPose (Chen et al., 2025) and CNOS (Nguyen et al., 2023) were among the first notable ones solving it. The core architecture of the latter, using template views as references and a SAM variant for proposal creation while classifying these via a similarity-based image matching technique, has laid the foundation for subsequent models such as SAM-6D and NIDS-Net. These works inspired us to adopt said approach as our starting point.

## 3 METHOD

In this section, we explain our approach for performing the instance segmentation, i.e. generating segmentation masks and labeling them, for all novel objects within an RGB query image $I \in \mathbb{R}^{3 \times W \times H}$, given just a set of RGB template images of said objects and without any (re-)training; where W and H are the width and height in pixels, respectively, and 3 is the number of the channels (RGB).

Our approach, as shown in Figure 1, is carried out in three steps. Starting with the onboarding stage in Section 3.1, visual descriptors are extracted from the template images via DINOv2; followed by the proposal stage in Section 3.2, where all possible segmentation masks and their descriptors, from the query RGB image, are generated with GSAM 2 and DINOv2, respectively. Lastly, in Section 3.3, the matching stage, each proposed mask is given an object label and a confidence value, based on the determined object scores using the visual descriptors.

### 3.1 ONBOARDING STAGE

The goal of the onboarding stage is to generate multiple visual descriptors to represent each of the $N^O$ different novel objects $\mathbb{O}$. In the following, in all the descriptions and notations, we will consider just one object $O \in \mathbb{O}$; this is done to keep the notation simple. In detail, the object $O$ is represented by a

set of $N^T$ template images $\mathbb{T}$ ($\mathbb{R}^{3 \times W' \times H'}$ images) and their corresponding ground truth segmentation masks showing the object from different predefined viewpoints. Given some fixed viewpoints, there are multiple possible sources for these templates and masks, e.g. pre-render them with renderers like Pyrender (Matthew Matl, 2021) or BlenderProc (Denninger et al., 2023); or even extract them out of some selected frames, e.g. annotated videos, where the object is not too occluded and has a viewpoint close to a predefined one.

In a preprocessing step, the segmentation masks are used to remove the background and to crop the object instance in each template; then, the crop size is unified via resizing and padding. Afterwards, the instance crops are fed into DINOv2 creating a class embedding/*cls* token $\boldsymbol{f}_T^{cls} \in \mathbb{R}^{N_{cls}^{dim}}$ and $N_T^{crop}$ patch embeddings/*patch* tokens $\boldsymbol{F}_T^{patch} = [\boldsymbol{f}_1^{patch} | \dots | \boldsymbol{f}_{N_T^{crop}}^{patch}] \in \mathbb{R}^{N_T^{crop} \times N_{patch}^{dim}}$ for each template $\boldsymbol{T} \in \mathbb{T}$, where $N_T^{crop}$ denotes the number of not masked out patches within the cropped template mask ($N_T^{crop} \leq N^{patch}$). The cropped templates are internally divided into $N^{patch} = 256$ patches, on a $16 \times 16$ grid, for the *patch* tokens. The *cls* token and *patch* tokens, together, form the visual descriptor of each template.

## 3.2 PROPOSAL STAGE

At this stage, all object proposals from the query image $\boldsymbol{I}$ are acquired. While previous works (Li et al., 2023; Shen et al., 2023; Chen et al., 2025; Nguyen et al., 2023; Lin et al., 2024) have employed various "pure" SAM-based proposal generators, we instead adopt GSAM 2 as a modified version of GSAM used in NIDS-Net. The original GSAM obtains the bounding boxes of all objects from Grounding-DINO (Liu et al., 2024), a pre-trained zero-shot detector, matching a given text prompt; then, it uses these as a prompt for SAM to create masks. GSAM 2 replaced its SAM component with the qualitative (w.r.t. mIoU) and performance-wise improved SAM 2.

Therefore, GSAM 2, with the text prompt "objects", is applied to the query image to extract all foreground object proposals $\mathbb{P}$. Each of the $N_I^{prop}$ proposals $p \in \mathbb{P}$ consists of a bounding box, a corresponding segmentation mask and a confidence score for both of them; note that $N_I^{prop}$ changes according to $\boldsymbol{I}$. All proposals whose confidence scores are lower than a threshold value or are too small, relative to the image size, are filtered out. Using the pipeline from the previous section, for each proposal $p$ the preprocessing step creates the image crop $\boldsymbol{I}_p$, which is then used by DINOv2 to generate the *cls* token $\boldsymbol{f}_{\boldsymbol{I}_p}^{cls}$ and *patch* tokens $\boldsymbol{F}_{\boldsymbol{I}_p}^{patch}$; which form the visual descriptors for all proposals.

## 3.3 MATCHING STAGE

At the matching stage, we calculate for each proposal-object pair their corresponding matching score, using the previously gathered visual descriptors; then, we assign to each proposal the most fitting object label and a confidence score.

The object matching score $s_p^{obj}$, between a proposal $p$ and an object $O$, represented by its templates $\mathbb{T}$, is made of different components; namely: the semantic score $s_p^{sem}$; the appearance score $s_p^{appe}$; and a proposal confidence $conf_p$.

**Semantic score** The semantic score $s_p^{sem}$ is used as a robust baseline measure of similarity via semantic matching and is defined as the top-5 average of the $N_T$ cosine similarity values between $\boldsymbol{f}_{\boldsymbol{I}_p}^{cls}$ and $\boldsymbol{f}_T^{cls}$ for all $\boldsymbol{T} \in \mathbb{T}$ *cls* tokens, where the cosine similarity is defined as:

$$cossim(\boldsymbol{a}, \boldsymbol{b}) = \frac{\langle \boldsymbol{a}, \boldsymbol{b} \rangle}{\|\boldsymbol{a}\| \cdot \|\boldsymbol{b}\|}, \tag{1}$$

with $\langle , \rangle$ denoting the inner product and $\| \cdot \|$ the Euclidean norm. If the vectors point in the same direction, they have a *cossim* value of $1$, $-1$ for opposite directions and $0$ for orthogonality. It was shown in CNOS (Nguyen et al., 2023, Section 4.3), that for this score, using the top-5 average as an aggregating function, is the most robust option out of: Mean; Max; Median and top-K Average.

**Appearance score with cyclic threshold** The semantic score alone represents a degree of similarity between the templates and a specific query object instance. Whenever two images show the same object, albeit with different viewpoints on it, this score should be high. Conversely, when two images

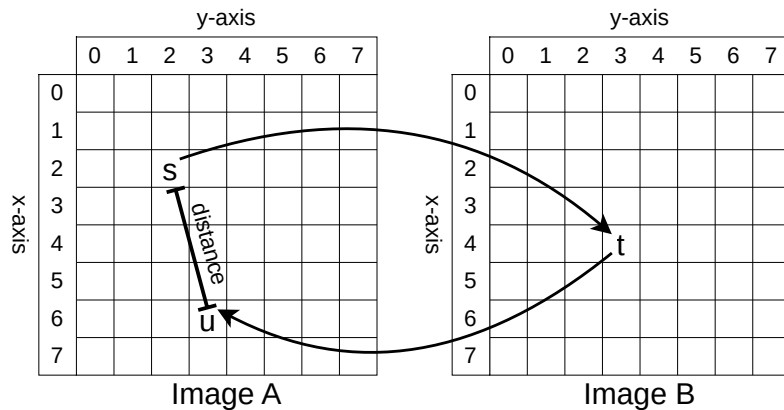

Figure 2: A general representation of the *cyclic* distance of patch $s$ through $t$ and $u$. Each image is divided into a $8 \times 8$ grid. Starting from patch $s$ in image $\boldsymbol{A}$, the most cosine similar patch in $\boldsymbol{B}$ is $t$. Vice versa, starting from $t$, its best match in $\boldsymbol{A}$ is $u$. The patch $u$ is the *cyclic*/round-trip patch of $s$, their euclidean distance is called the *cyclic* distance.

are showing different objects, their similarity score should drop. However, there might be cases where two different objects, despite their different appearances, are still semantically similar to each other (e.g. two food cans). To address this issue, it is necessary to introduce the concept of an appearance score $s_p^{appe}$, which gives a way to discriminate between objects which are semantically similar, but with different patch/part-wise appearance. Indeed, one can consider the average of the best possible semantic scores for each proposal patch against all template patches using their respective *patch* tokens as a way to define it. Starting with each of the $N_T$ proposal-template pairs, a sub-appearance score $s_{p,\boldsymbol{T}}^{appe}$ is assigned for template $\boldsymbol{T} \in \mathbb{T}$, which is defined as follows:

$$s_{p,\boldsymbol{T}}^{appe} = \frac{1}{N_{\boldsymbol{I}_p}^{crop}} \sum_{i=1}^{N_{\boldsymbol{I}_p}^{crop}} \max_{j=1,...,N_{\boldsymbol{T}}^{crop}} \left( cossim(\boldsymbol{F}_{\boldsymbol{I}_p,:,i}^{patch}, \boldsymbol{F}_{\boldsymbol{T},:,j}^{patch}) \right) \cdot \mathbf{1}_{cdist(\boldsymbol{I}_p, \boldsymbol{T}, i) \leq \delta_{CT}}, \qquad (2)$$

where $\boldsymbol{F}_{\boldsymbol{I}_p,:,i}^{patch}$ represents the $i$-th column of the *patch* token matrix $\boldsymbol{F}_{\boldsymbol{I}_p}^{patch}$; $\boldsymbol{F}_{\boldsymbol{T},:,j}^{patch}$ the $j$-th column of $\boldsymbol{F}_{\boldsymbol{T}}^{patch}$; and $\mathbf{1}$ the indicator function, which is used to filter out certain patch pairs. As these sub-appearance scores show an inherent sensitivity to the visible parts of the object instances, thus to the viewpoint differences, they are aggregated into the final appearance score $s_p^{appe}$ using the Max function (across the templates), to mitigate this phenomenon.

The idea behind the *cyclic* threshold (CT) filtering arises as DINOv2 descriptors can assign similar *patch* tokens to repetitive textures/similar looking parts (e.g. identical corners or surfaces), leading to many-to-one matches, which one would like to avoid via patch filtering of sorts. An often used technique to solve this issue is the nearest neighbor based image patch matching, e.g. for finding the most coherent pairs of patches Simakov et al. (2008) and points Oron et al. (2018); however, in practice, this incurs in a restrictive filtering, that is why we relaxed this bidirectional similarity aspect to account for a non strictly one-to-one mapping assumption between the template and the query regions, as different scene lighting and occlusions might dampen it. Therefore, our CT filtering allows a matching that is not just strictly mutual but permits a certain degree of tolerance, i.e. how many patches in the neighborhood of the considered ones, in terms of Euclidean distance, are still accepted.

In Figure 2, the general representation of the *cyclic* distance of patch $s$, through $t$ and $u$, is given; where $s$ and $u$ belong to image $\boldsymbol{A}$ and $t$ to $\boldsymbol{B}$. Each example image is divided into an $8 \times 8$ grid for the sake of simplicity, resulting in $64$ patches. From patch $s$ in image $\boldsymbol{A}$, we find $t$, the most similar patch of it in $\boldsymbol{B}$, via the following function:

$$bestMatchIndex(\boldsymbol{F}_{\boldsymbol{A}}^{patch}, \boldsymbol{F}_{\boldsymbol{B}}^{patch}, i) = \operatorname*{argmax}_{j=1,...,N_{\boldsymbol{B}}^{crop}} cossim(\boldsymbol{F}_{\boldsymbol{A},:,i}^{patch}, \boldsymbol{F}_{\boldsymbol{B},:,j}^{patch}), \qquad (3)$$

where $t = bestMatchIndex(\mathbf{F}_A^{patch}, \mathbf{F}_B^{patch}, s)$ using the DINOv2 *patch* tokens of $\mathbf{A}$ and $\mathbf{B}$. Vice versa, starting from $t$, its best match in $\mathbf{A}$ is patch $u$; by using the same function. We, therefore, call $u$ the *cyclic*/round-trip patch of $s$ and their euclidean distance, on the grid, the *cyclic* distance of $s$, namely $cdist$.

Given the previous discussion regarding the mutual similarity principle, using a CT value of $0$ is equivalent to mutual nearest neighbor based image patch matching. Therefore using the previously defined *cyclic* distance, our addition to the appearance score lies in the application of a patch filter, represented by the indicator function $\mathbf{1}$ in equation 2, to increase the score's reliability/expressiveness; which allows for a relaxed mutual nearest neighbours matching. The function $\mathbf{1}_{cdist(\mathbf{I}_p, \mathbf{T}, i) \leq \delta_{CT}}$ internally calculates the $cdist$ for patch $i$ of image crop $\mathbf{I}_p$ and template $\mathbf{T}$, then checks if it is smaller than a predefined CT value. The default value for it is $\delta_{CT} = 5$, see also the Appendix Section A.1, where the effects of using different threshold values are discussed.

Lastly, to sum up our contributions to the appearance score, one can see that it significantly differs from the approach used in SAM-6D (Lin et al., 2024, Section 3.1.2), as its authors computed only the sub-appearance score for the single template of the object having the highest semantic score, thus resulting in a highly biased score. Indeed, as it was shown in CNOS (Nguyen et al., 2023, Section 4.4), the *cls* token contains insufficient information about matching viewpoints, potentially leading to low appearance values. Additionally, we included our CT filtering technique to improve the appearance score accuracy even further.

**Bounding box and segmentation mask confidence** Proposals might contain a high number of false positives; indeed, background regions and object parts might be misinterpreted as complete objects. To account for this, for each proposal $p$, the proposal confidence $conf_p$, as the average confidence value of its bounding box and segmentation mask, is included as a weighting factor for the object matching score in the next paragraph.

**Object matching score** By combining the previously mentioned scores and the proposal's confidence, the object matching score $s_p^{obj}$ is determined as follows:

$$s_p^{obj} = \frac{s_p^{sem} + w_{appe} \cdot s_p^{appe}}{1 + 1} \cdot conf_p, \tag{4}$$

where an appearance weight of $w_{appe} = 1$ computes the average. The object matching scores of all the $N_I^{prop}$ proposals, over all possible $N^O$ objects, are stored in the $N_I^{prop} \times N^O$ instance score matrix. Note that, as small CT values squash down the appearance scores, a $w_{appe} = 2$ is used.

**Object label assignment** In the last step, we simply apply the Argmax function across the objects/rows of the instance score matrix. Each proposal gets assigned an object label and its object matching score as its corresponding confidence. Eventually, we obtain proposals consisting of: a bounding box of the object instance; its corresponding modal segmentation mask, which covers the visible instance part (Hodan et al., 2024); and an object label with a confidence score. To remove incorrectly labeled proposals and redundant ones, a confidence threshold filtering is applied with $\delta_{conf} = 0.2$ followed by a Non-Maximum Suppression, respectively.

## 4 EXPERIMENTS

In this section we first present our experimental setup (Section 4.1), followed by a comparison of our method with the SOTA ones, across the seven core datasets of the BOP 2023 challenge (Section 4.2). Moreover, we perform ablation studies regarding the score components' choices in Section 4.3. Finally, in the last Section 4.4, we discuss some limitations of NOCTIS.

### 4.1 EXPERIMENTAL SETUP

**Datasets** We evaluate our method on the seven core datasets of the BOP 2023 challenge: LineMod Occlusion (LM-O) (Brachmann et al., 2014); T-LESS (Hodan et al., 2017); TUD-L (Hodan et al., 2018); IC-BIN (Doumanoglou et al., 2016); ITODD (Drost et al., 2017); HomebrewedDB (HB) (Kaskman et al., 2019) and YCB-Video (YCB-V) (Xiang et al., 2018). Overall those datasets contain 132 household and industrial objects, being textured or untextured, and are symmetric or

asymmetric; moreover, they are shown in multiple cluttered scenes with varying occlusion and lighting conditions.

**Evaluation metric**    As evaluation criterion for the "2D instance segmentation of unseen objects" task, we use the Average Precision (AP) following the standard protocol from the BOP 2023 challenge. The AP metric is computed as the average of precision scores, at different Intersection over Union (IoU) thresholds, in the interval from 0.5 to 0.95 with steps of 0.05.

**Implementation details**    To generate the proposals, we use GSAM 2, with an input text prompt "objects", comprised of the Grounding-DINO model with checkpoint "Swin-B" and SAM 2 with checkpoint "sam2.1-L". The corresponding regions of interest (ROIs) are resized to $224 \times 224$, while using padding to keep the original size ratios. We use the default "ViT-L" model/checkpoint of DINOv2 (Oquab et al., 2024), for better comparability with previous works Nguyen et al. (2023); Lin et al. (2024); Lu et al. (2025), to extract the visual descriptors as 1024-dimensional feature vectors ($N_{cls}^{dim} = N_{patch}^{dim} = 1024$), where each *patch* token on the $16 \times 16$ grid represents $14 \times 14$ pixels. We use the "PBR-BlenderProc4BOP" pipeline with the same 42 predefined viewpoints, as described in CNOS (Nguyen et al., 2023, Sections 3.1 and 4.1), to select the templates representing every dataset object. See also the Appendix Section A.1, for a quick comparison between different types of template renderers.

The main code is implemented in Python 3.8 using Numpy (Harris et al., 2020) and PyTorchPaszke et al. (2019) (Version 2.2.1 CUDA 11.8). To ensure reproducibility, the seed values of all the (pseudo-) random number generators are set to 2025. The tests were performed on a single Nvidia RTX 4070 12GB graphics card and the average measured time per run, with one run using the same configuration on all the seven datasets, was approximately 90 minutes or 0.990 seconds per image.

## 4.2 COMPARISON WITH THE STATE-OF-THE-ART

We compare our method with the best available results from the leaderboard[2] of the BOP challenges, comprising of the top-3 paper-supported methods: CNOS, SAM-6D and NIDS-Net; and the overall top-3 ones: "anonymity", LDSeg and MUSE (November 2024 and July 2025 version); which do not have a paper or code publicly available. CNOS uses proposals from SAM or FastSAM and only the semantic score 3.3 for matching. SAM-6D uses the same proposals and semantic score as CNOS; additionally, it uses an appearance-based and geometric matching score of the single template with the highest semantic score; the latter score utilizes depth information to consider the shapes and sizes of instances during matching. NIDS-Net uses proposals from GSAM and the similarity between the weight adapter refined Foreground Feature Averaging (Kotar et al., 2023) embeddings together with SAM-6D's appearance score. For the methods "anonymity", LDSeg and MUSE, no further information nor clear details are available.

In the Table 1 we show the results for NOCTIS and the other methods on all seven datasets and the overall average. We surpass the best established method NIDS-Net by a significant margin of absolute 3.4% mean AP and we are on par with the best one (MUSE, July 2025 version); overcoming the second best undisclosed one by 0.8%. To further highlight that our pipeline performs better (in terms of mean AP score) than previous works like NIDS-Net, and that its results do not only stem out of a better foundation model; GSAM 2 was replaced with the older GSAM. The results in the last line of Table 1 show that, even when one makes this change, NOCTIS still achieves results comparable to the SOTA by showing an overall mean AP score of 0.513. Notably, our methodology outperforms the ones that are using depth data as well (see Section 4.4).

 In Figure 3 we show some qualitative segmentation results of our method vs. the publicly available ones, where errors in the masks and/or classifications of the proposals are indicated by red arrows. One can clearly see that all the methods have their own strengths and weaknesses. CNOS and SAM-6D, for example, as they are using SAM/FastSAM as a proposal generator, have problems in differentiating between the objects and some of their parts. While NIDS-Net, due to its internal usage of GSAM, does not suffer from the previously mentioned problem, it still produces misclassifications in the form of labeling scene objects wrongly or by producing oversized bounding boxes around correctly identified objects, leading to multiple detections. NOCTIS, on the other hand, suffers less from said problems, but it is still prone to misclassification, like the other methods, when the objects are too similar looking or too close to each other; see columns 4 (left clamp) and 6 (bottom

---

[2]https://bop.felk.cvut.cz/leaderboards/segmentation-unseen-bop23/bop-classic-core/; Accessed: 2025-07-28

Table 1: Comparison of NOCTIS against different methods on the seven core datasets of the BOP 2023 challenge, w.r.t. the AP metric (higher is better). For each dataset, the best result is displayed in bold and the second best is underlined. NOCTIS(*) uses GSAM instead of GSAM 2.

| Method | Depth | BOP Datasets | | | | | | | Mean |
|---|---|---|---|---|---|---|---|---|---|
| | | LMO | TLESS | TUDL | ICBIN | ITODD | HB | YCBV | |
| CNOS | - | 0.397 | 0.374 | 0.480 | 0.270 | 0.254 | 0.511 | 0.599 | 0.412 |
| SAM-6D | ✓ | 0.460 | 0.451 | 0.569 | 0.357 | 0.332 | 0.593 | 0.605 | 0.481 |
| NIDS-Net | - | 0.439 | **0.496** | 0.556 | 0.328 | 0.315 | 0.620 | 0.650 | 0.486 |
| MUSE | - | 0.478 | 0.451 | 0.565 | 0.375 | **0.399** | 0.597 | 0.672 | 0.505 |
| LDSeg | ✓ | 0.478 | 0.488 | **0.587** | 0.389 | 0.370 | 0.622 | 0.647 | 0.512 |
| anonymity | - | 0.471 | 0.464 | 0.569 | 0.386 | 0.376 | 0.628 | 0.688 | 0.512 |
| MUSE(new) | - | 0.476 | 0.486 | 0.550 | **0.408** | 0.382 | **0.636** | **0.702** | **0.520** |
| NOCTIS | - | **0.489** | 0.479 | 0.583 | 0.406 | 0.389 | 0.607 | 0.684 | **0.520** |
| NOCTIS(*) | - | 0.484 | 0.483 | 0.567 | 0.391 | 0.386 | 0.613 | 0.664 | 0.513 |

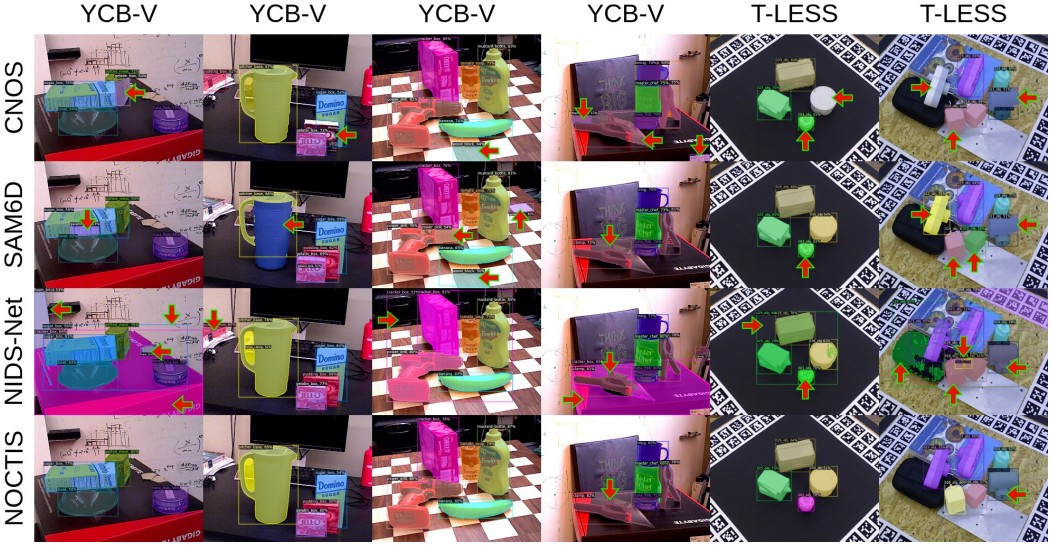

Figure 3: Qualitative assessment of some segmentation results using CNOS, SAM-6D, NIDS-Net, and NOCTIS on YCB-V and T-LESS. The image addresses the strengths and limitations of these methods. The red arrows indicate errors in the segmentation masks and/or classifications of the proposals. For better visualization purposes, $\delta_{conf} = 0.5$ was used.

electric boxes and right adapters) for reference. Indeed, in the YCB-V dataset, due to the high similarity between the different sized clamps, one of them is easily confused with the other; the same is true for the electric boxes in T-LESS. Besides, the two stacked adapters are confused as one object and correspondingly misclassified. Overall, NOCTIS still displays fewer errors, on average, when compared to the other methods.

## 4.3 ABLATION STUDIES

**Score components** In lines $1-6$ of Table 2, we show the influence of the different score components on the mean AP metric to justify our architecture. Line 0 shows the result attained by the complete NOCTIS model as shown in Table 1. The combined use of semantic and appearance scores leads to a better mean AP score than just using them alone, as shown in lines $1-3$. Line 4 then shows that the addition of the proposal confidence to the semantic score also increases the performances significantly.

Table 2: Ablation studies on the influence of the components on the mean AP metric. In the $s^{appe}$ column, the values represent the corresponding value of $w_{appe}$, as shown in equation 4.

|   | $s^{sem}$ | $s^{appe}$ | CT Filter | $conf$ | Mean |
|---|---|---|---|---|---|
| 0 | ✓ | 2 | ✓ | ✓ | **0.520** |
| 1 | ✓ | - | - | - | 0.464 |
| 2 | - | 1 | - | - | 0.480 |
| 3 | ✓ | 1 | - | - | 0.494 |
| 4 | ✓ | - | - | ✓ | 0.494 |
| 5 | ✓ | 1 | - | ✓ | 0.512 |
| 6 | ✓ | 1 | ✓ | ✓ | 0.516 |

Furthermore, in a pure incremental fashion, one can see that adding the confidence to the pipeline in line 3, resulting in line 5, and on top of that adding the CT filter with a value of $\delta_{CT} = 5$ yields better results, see lines 5 and 6. As one can easily notice, the addition of the score components causes some clear performance gains; however, one cannot justify the inclusion of certain score components in our pipeline by the sheer increase in mean absolute AP score they provide on their own, as one is bound to meet diminishing returns at some point. Intuitively, getting a better mean AP score over all datasets is a daunting task, as some changes in the pipeline might result in better gains over some of them, but decreases over others. As a side note, line 1, when compared to the standard CNOS (see Table 1), emphasizes the importance of proposal mask quality; thus validating our proposal generator of choice, GSAM 2, against the previously used ones, i.e. SAM and FastSAM.

Further ablation studies regarding different CT values; the effect of $w_{appe}$ on the mean AP score; and the usage of different renderers can be found in the Appendix Section A.1.

### 4.4 LIMITATIONS

As seen in column 4 from Figure 3, our method does not perform at its highest when the objects are similar looking but different sized, e.g. all the clamps from YCB-V; or they are untextured (see column 6), e.g. the industrial models from ITODD. While these issues might be solved by using depth data, it does not seem that easy, as e.g. SAM-6D is still not able to solve this issue.

Further discussion and limitations regarding the memory usage and runtime are present in the Appendix Section A.2.

### 5 CONCLUSION

In this paper we presented NOCTIS, a new framework for zero-shot novel object instance segmentation; which leverages the foundation models Grounded-SAM 2 for object proposal generation and DINOv2 for visual descriptor based matching scores. The novelties introduced in Section 3 have proven to be largely effective; indeed, NOCTIS was able to perform better in terms of mean AP than all the other methodologies (barring MUSE July 2025 version), disclosed or not, on the seven core datasets of the BOP 2023 benchmark. This shows that it is not necessary to have overly complicated scores to achieve good performances. We hope that our work can be used as a new standard baseline to improve upon, especially regarding the formulation of a better scoring rule.

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

# A  APPENDIX

## A.1  FURTHER ABLATION STUDIES

**Template creation**    Table 3 demonstrates the effects of using different template sources on the mean AP metric. Line 1 refers to Pyrender as a lightweight/fast renderer (similar to CNOS) and line 2 to BlenderProc for more photo realistic renders; which was utilized by GigaPose (Nguyen et al., 2024) for their "onboarding" stage. Both render the object floating in an empty (black) space.

Our default "PBR-BlenderProc4BOP" pipeline gave the best results; while it also uses BlenderProc for rendering, it renders objects in random cluttered scenes to make the reference crops even more realistic. Indeed, the crops obtained this way appear more realistic because of the cross-effects of other objects being in the scene (e.g. slight partial occlusions, other objects' shadows, changes in the ambient light perspective), compared to standalone objects that produce too clean/artificial templates that seem to have a negative influence on the matching.

For example, if one were to consider the same object in two different scenes, one without any other object and the other with an object that is slightly occluding a part of it. The resulting masks would be different, as the one in the "void" would result in a representation of the object that shows all of its parts; the same cannot be said about the other case, as some of its parts would not be present in its mask. As it turns out, these occluded objects can better represent the ones in the cluttered scenes of the query/test images of the BOP challenge.

Table 3: Ablation study on the influence of the adopted template creation technique on the mean AP metric.

|   | Renderer | Mean |
|---|----------|------|
| 0 | PBR | **0.520** |
| 1 | Pyrender | 0.482 |
| 2 | Blender | 0.509 |

**Varying cyclic threshold**    Table 4 exhibits the effects of utilizing various cyclic thresholds on the mean AP metric. Smaller CT values filter out too many patches, thus reducing the performances. On the other hand, too large ones are prone to noise. A CT value of $5$ seems to be the point after which the performances drop.

As it can easily be noticed, Table 4 only shows results for fixed values of the CT used across all objects of all datasets; thus one might wonder what would be the effects of using values adapted to each object, e.g. based on their texture complexity or viewpoint variation, rather than having only a static one. While it might look beneficial to adapt the CT value on object-specific characteristics, NOCTIS is a zero-shot pipeline requiring no further tuning, aligning with BOP's challenge goals of methods that work out-of-the-box on novel objects. Introducing adaptive thresholds (partly) undermines this zero-tuning philosophy and adds dataset-specific hyperparameters that would require further tuning, probably leading into loss of generalization on new object instances. Moreover, the adjusted CT value of an object would most likely be affected by cross influences of the other objects, making NOCTIS incapable of adding new objects on-the-fly, as re-adjusting these values would always be needed. Eventually, it is unclear, from the get-go, on how many different scenarios one would have to test these new varying CT values internally, considering also the required runtime and memory for such an optimization, to obtain a reliable, general and easily scalable model.

**Appearance score weight**    Table 5 illustrates how different $w_{appe}$ values affect the mean AP score on two different configurations of our pipeline; where the first one uses only the semantic and appearance score, while the second one represents the full/best pipeline. While the choice of including or not the appearance score is influential (see Section 4.3), changing its weight as shown by the above results does not provide significant gains/losses over the final performances.

As a side note, the value $w_{appe} = 2$ was originally chosen because some experiments with $\delta_{CT} = 0$ highlighted that the appearance score of some desired object was (roughly) halved, while for the undesired ones it was reduced to (roughly) a third. To compensate for this effect, those values were

Table 4: Ablation study on the influence of different cyclic threshold (CT) values on the mean AP metric.

| CT | Mean |
|----|------|
| 0 | 0.479 |
| 1 | 0.500 |
| 2 | 0.512 |
| 3 | 0.517 |
| 4 | 0.518 |
| 5 | **0.520** |
| 6 | 0.517 |
| 7 | 0.517 |
| 8 | 0.516 |

Table 5: Ablation study on the influence of different $w_{appe}$ values on the mean AP metric.

| $w_{appe}$ | $s^{sem} + s^{appe}$ | Full |
|------------|----------------------|------|
| 1 | 0.494 | 0.516 |
| 2 | **0.497** | **0.520** |
| 3 | 0.495 | 0.517 |
| 4 | 0.493 | 0.514 |

doubled to have the same level of magnitude as they would have had before; otherwise, they would make the overall object score too small, risking filtering out of the proposal at the end.

## A.2 ADDITIONAL DISCUSSION AND LIMITATIONS

**Memory usage and runtime** The memory usage for determining object matching scores, for a query image, has two bottlenecks; each of them requires calculating a large matrix at some point. A $N_I^{prop} \times N^O \times N^T$ matrix for the semantic scores before the aggregation, like in CNOS; and a "big" $N_I^{prop} \times N^O \times N^T \times N^{patch} \times N^{patch}$ internal floating-point matrix for the appearance scores. As a side note, one could speculate that the size of the latter matrix could be the reason why SAM-6D, originally, computed the appearance score only for one template. *Cyclic* filtering only temporarily adds ca. seven $N_I^{prop} \times N^O \times N^T \times N^{patch}$ matrices ($5 \times integer + 1 \times float + 1 \times bool$) to the memory, which is ca. $36\times$ less memory compared to the big one and so negligible.

To reduce the CUDA memory requirements, we make use of mini batches on the $N_P$ and $N_O$ dimensions; however, this comes with an increase in computation time. Furthermore, until the templates remain the same between different runs; their visual descriptors stay the same after the onboarding stage (see Section 3.1), since DINOv2 is deterministic, they can be stored and cheaply reloaded later.

Regarding the runtime, the original full NOCTIS pipeline needs 0.990 seconds per image, while the other configuration, without CT filtering (Table 2 line 5), requires 0.995 seconds. Given that the evaluation was run on a normal office desktop PC, the difference is minimal and can be seen as noise. This short analysis, contrarily to what one might intuitively assume, shows that using the CT component in our pipeline does not imply heavy additional memory and runtime requirements.

**Other BOP datasets** Our method is only evaluated on the seven core BOP 2023 datasets; this is done due to, on one hand, a lack of other results' data for the BOP classic Extra datasets (LM (Hinterstoisser et al., 2013); HOPEv1 (Tyree et al., 2022); RU-APC (Rennie et al., 2016); IC-MI (Tejani et al., 2014) and TYO-L (Hodan et al., 2018)); on the other hand, because the BOP 2024 (Nguyen et al., 2025) and 2025 (Tomas Hodan & Fourmy, 2025) challenges are targeting only a detection task. But, as mentioned in Section 4.1, the chosen datasets provide a wide range of different scenes; therefore, their evaluation should still be reliable.

### A.3 ADDITIONAL COMMENTS

**Zero-shot generalization**  Zero-shot generalization has been one of the main driving forces behind NOCTIS' implementation. Moreover, this is an important prerequisite for the "Model-based 2D segmentation of unseen objects" task of the BOP challenge to have. Indeed, NOCTIS just needs some template views (see Section 3.1) for representing any object from any kind of image source (renderer, video frames, hand-made camera images, etc). For example, one could just take their camera to shoot some photos of an (rigid) object from multiple viewpoints, mask out the object (somehow) and feed the masked photos to NOCTIS as template views; afterwards the object can be detected. Thus, both the CAD/3D object models and the fixed viewpoints are not really necessary for our pipeline; as the models are only needed when rendering the object as a template source, while the viewpoints ensure a good overview of the object. Correspondingly, they are relevant only in the context of the BOP challenge for the evaluation of our pipeline according to the provided benchmarks; which means applying NOCTIS outside of the BOP benchmarks can be easily achieved.

As each object is represented by some embeddings/tokens stored in memory, and there are no cross-influences among tokens belonging to different objects, it is possible to manipulate the memory to allow for the addition/removal of objects on-the-fly. This aspect is relevant for example in industrial settings where one needs to dynamically detect objects without suffering from onboarding downtime; indeed, a whole database of possible objects can be precomputed and stored, making it easy to load or unload only the required ones from it.

**BOP challenge runtime**  While computational efficiency is usually an important factor for real world applications, it is not an evaluation criterion for the BOP task. Given the fact that different hardware configurations heavily influence the runtime ("normal" graphic cards vs. server ones), the total time per image as stated in the BOP challenge rules is not a major factor regarding the quality of the algorithms proposed. Yet, our pipeline would benefit from better hardware/more CUDA memory, as the "Nvidia RTX 4070 12GB" graphic card used for the experiments is just a standard graphic card compared to what the other participants for the same BOP task were using, like e.g. "GeForce RTX 3090 24GB" (SAM-6D) or "V100 16GB" (CNOS).

