# OpenReview forum: "NOCTIS: Novel Object Cyclic Threshold based Instance Segmentation"
_ICLR.cc/2026/Conference — ICLR 2026 Conference Withdrawn Submission_

### Official Review · Reviewer_BfXR · 2025-10-18

**Soundness:** 2
**Presentation:** 2
**Contribution:** 2
**Rating:** 4
**Confidence:** 3

**Summary:**

This paper introduces NOCTIS, an instance segmentation framework that combines pre-trained large models with a proposed cyclic threshold. To perform instance segmentation, the framework first takes multi-view template images and masks that depict the objects of interest, and encode them as template features for reference. Afterwards, the framework takes an image as input and gets proposals from GSAM2, the proposals are then matched with the template features using the object matching score (which involves the cyclic threshold) to get the final detection & segmentation results.

**Strengths:**

1. The proposed approach is straightforward and not difficult to understand.
2. Experiments on NOCTIS yield impressive results.

**Weaknesses:**

1. The overall contribution, which I believe centers around Eq.4, appears limited for an ICLR submission.
2. The claimed contribution on "removing selection bias" is not supported by experiments.
3. The proposed approach introduces additional parameters such as CT and $w_{appe}$, which adds to the difficulties in parameter tuning for real-world usage.
4. (Minor) It appears to me that the paper's choice of language style is more like a speech than a research paper.

**Questions:**

1. Is it true that for real-world usages, one must possess a 3D model or at least several multi-view images of the target object before the proposed framework can function properly? If so, this may significantly add to the effort for deployment.
2. The proposed cyclic distance uses the Euclidean Distance on images, does this mean that NOCTIS naturally performs differently for the same object when it is closer (appears bigger) or farther away (appears smaller) from the camera?
3. How is the inference time of NOCTIS compared with other baseline methods?

---

### Official Review · Reviewer_boL7 · 2025-10-30

**Soundness:** 3
**Presentation:** 2
**Contribution:** 3
**Rating:** 4
**Confidence:** 3

**Summary:**

This paper introduces NOCTIS, a new training-free framework for instance segmentation of novel objects. It works by integrating two powerful pre-trained models: Grounded-SAM 2 (GSAM 2) to generate high-quality object proposals, and DINOv2 to extract robust visual features.
The core innovations of NOCTIS are found in its matching algorithm. It proposes a novel "Cyclic Thresholding" (CT) mechanism to solve matching instability caused by repetitive textures. Furthermore, it incorporates an unbiased appearance score and, for the first time, uses the "proposal confidence" from GSAM 2 as a weight in the final matching score.
As a result, NOCTIS achieves state-of-the-art (SOTA) performance on the seven core datasets of the BOP 2023 challenge. Notably, it accomplishes this using only RGB images, surpassing the performance of previous SOTA methods like NIDS-Net and SAM-6D, which rely on RGB-D (depth) data.

**Strengths:**

1. SOTA on RGB-Only: Achieved SOTA on the BOP benchmark using only RGB images, outperforming methods that rely on RGB-D (depth) data.
2. Novel CT Matching Algorithm: This paper introduced the "Cyclic Thresholding" (CT) mechanism, a new and effective algorithm that addresses DINOv2's matching instability on repetitive textures.

**Weaknesses:**

First, the paper's core premise of "novelty" is questionable. The framework relies heavily on foundation models (GSAM 2 and DINOv2) that were pre-trained on massive datasets. It is highly probable that these models have already "seen" the object categories present in the BOP benchmark. Therefore, the "zero-shot" capability claimed is more a feat of the models' generalization than true segmentation of unseen objects.

Second, the innovation is incremental and best described as a clever engineering assembly rather than a fundamental advancement. The method is essentially a "glue" for existing model outputs, relying on a series of manually-tuned heuristics and hyperparameters. This questions the generalizability of the solution, as it appears more like a "custom-fit" for the BOP benchmark than a robust, general-purpose method.

Finally, the system suffers from structural fragility and severe efficiency bottlenecks. Its performance is entirely capped by the quality of the upstream proposal generator (GSAM 2); any missed detection or poor segmentation is unrecoverable. Moreover, the paper notes considerable memory bottlenecks, as the computational load scales significantly with the number of proposals and objects, which may present scalability challenges for real-world scenarios.

**Questions:**

See weakness above

---

### Official Review · Reviewer_5ifb · 2025-11-01

**Soundness:** 2
**Presentation:** 2
**Contribution:** 2
**Rating:** 4
**Confidence:** 2

**Summary:**

This paper presents a novel training-free and RGB-only framework for zero-shot novel object instance segmentation that leverages the zero-shot capabilities of two pre-trained foundation models: Grounded-SAM 2 and DINOv2. The key contributions for reliable proposal-object matching are: (i) a Cyclic Thresholding (CT) mechanism, a novel patch-filtering strategy designed to mitigate unstable matches from repetitive textures; (ii) an unbiased appearance score that aggregates results over all object templates to remove selection bias ; and (iii) the incorporation of the proposal's mask and bounding-box confidence values into the final object matching score. Experiments show NOCTIS achieves state-of-the-art (SOTA) on the BOP 2023 challenge.

**Strengths:**

- Performance: This paper successfully integrates multiple methodological approaches, thereby translating the generalization capability of foundation models into State-of-the-Art performance in the BOP 2023 challenge.
- Reproducibility: The paper provides a large amount of detailed description about the experimental setup (e.g., software versions used, random seed, hardware configuration, and running time), which is very helpful for ensuring the good reproducibility of the results.
- One of the core proposed techniques, Cyclic Thresholding, is a robust patch matching mechanism based on DINOv2, which can be borrowed by other works using feature matching.

**Weaknesses:**

1. Lack of Novelty: It seems that the proposed method primarily relies on the integration of minor innovations (e.g., confidence and score aggregation) on top of existing foundation models. It is largely built upon prior works such as CNOS[1] and SAM-6D[2], resulting in limited incremental novelty.
2. Although the Cyclic Thresholding (CT) mechanism is highlighted as a major innovation, its actual performance gain is extremely limited (0.512 to 0.516 in ablation), which is disproportionate to the attention it receives in the paper.
3. The Introduction could be strengthened by more clearly identifying the key limitations of existing work to better clarify the motivation for the proposed method's innovations.
4. Writing. Within Section 3.3, "Appearance score with cyclic threshold," the two distinct contributions "unbiased appearance score" and the "cyclic threshold" lack clear separation, and the former is difficult to locate. We recommend that the authors use sub-headings or clearer structure to explicitly indicate the position of these two innovative points.

**Questions:**

1. Could the authors provide a visualization showing 'many-to-one' erroneous matches caused by repetitive textures (Line 055), and explain how the CT mechanism resolves this issue?
2. The data (e.g., the jump from 0.412 at Line 385 to 0.464 at Line 438) suggests that the foundation model upgrade (from SAM to Grounded-SAM 2) provides a significant performance boost (as said in Line 454). To ensure a fair comparison, we strongly recommend using Grounded-SAM 2 as the foundation model for baselines like SAM-6D[2] and NIDS-Net[3]. This will nullify the effect of the model upgrade and allow for an accurate quantification of the net gain from the paper's own components.
3. The NOCTIS technique, "An unbiased appearance score that aggregates over all templates," sounds very similar to the CNOS[1] (Sec 3.3) technique, "aggregating the similarity scores over all V templates." Could the authors explicitly state the key technical differences and advantages of the NOCTIS aggregation method compared to CNOS[1]?
4. What is the theoretical or experimental basis for claiming the 'unbiased' appearance score (Line 077)? Could the authors justify this strong assertion in the paper?

[1] Nguyen, V. N., Groueix, T., Ponimatkin, G., Lepetit, V., & Hodan, T. (2023). Cnos: A strong baseline for cad-based novel object segmentation. In Proceedings of the IEEE/CVF International Conference on Computer Vision (pp. 2134-2140).
[2] Lin, J., Liu, L., Lu, D., & Jia, K. (2024). Sam-6d: Segment anything model meets zero-shot 6d object pose estimation. In Proceedings of the IEEE/CVF Conference on Computer Vision and Pattern Recognition (pp. 27906-27916).
[3] Lu Y, Guo Y, Ruozzi N, et al. Adapting pre-trained vision models for novel instance detection and segmentation[J]. arXiv preprint arXiv:2405.17859, 2024.

---

### Official Review · Reviewer_5yiC · 2025-11-09

**Soundness:** 2
**Presentation:** 2
**Contribution:** 2
**Rating:** 4
**Confidence:** 4

**Summary:**

This paper introduces NOCTIS, a training-free, RGB-only framework designed for novel object instance segmentation, addressing the limitations of traditional methods that require costly retraining for new objects. The framework achieves state-of-the-art (SOTA) performance on the BOP 2023 benchmark. NOCTIS builds on foundation models (Grounded-SAM 2 and DINOv2) and introduces three main contributions: an unbiased appearance score that aggregates information over all templates to remove object selection bias; a cyclic thresholding mechanism that provides robust patch matching, especially for repetitive textures; and the pioneering incorporation of mask and bounding-box confidence scores into the final matching score.

**Strengths:**

1) Proposes a training-free method for novel class instance segmentation.

2) Proposes the cyclic thresholding method to mitigate the multi-to-one matching problem caused by strict matching.

3) Achieves SOTA or SOTA-comparable performance.

**Weaknesses:**

1) What is the relationship between the task defined in this paper and open-set/open vocabulary instance segmentation? The author claims it is infeasible to train an instance segmentor that can cover sufficiently many instances, yet this is precisely what open-set/open vocabulary instance segmentation tasks aim to achieve. These tasks also design their models based on the generalization capabilities of powerful pre-trained models like SAM and DINO.

2) The related work section also lacks a comparison with the open-set/open vocabulary line of methods.

3) Novelty: The paper introduces commonly used concepts such as foundation models and appearance scores, making the framework feel like a combination of existing techniques. While the cyclic thresholding method has some novelty, it might not be substantial enough to support a paper at ICLR.

**Questions:**

1) As a template matching-based method, how does it perform when objects are heavily occluded? How are occluded targets segmented correctly?

2) The weights in Equation 4 only used the parameters 1 and 2. The ablation experiments (e.g., Table 2) set the weight for $s^{appe}$ to 1, but the final model uses 2. How did the authors determine that 2 is the optimal parameter?

3) In Table 1, the proposed method NOCTIS achieves the same performance compared with MUSE(new). Is NOCTIS faster than MUSE(new) during inference?

---

### Note · Authors · 2025-11-12

I have read and agree with the venue's withdrawal policy on behalf of myself and my co-authors.